# Effect of Crystallographic Orientations on Bendability in a Strongly Textured Mg-9Al Extrusion Plate and Texture Evolution during Three-Point Bending

**DOI:** 10.3390/ma16062518

**Published:** 2023-03-22

**Authors:** Jiansheng Wei, Shunong Jiang, Yingchun Wan, Chuming Liu

**Affiliations:** 1School of Metallurgy and Environment, Central South University, Changsha 410083, China; weijiansheng@csu.edu.cn; 2School of Materials Science and Engineering, Central South University, Changsha 410083, China; 3Light Alloy Research Institute, Central South University, Changsha 410083, China; ycwanmse@csu.edu.cn; 4Hunan Meiyu Technology Co., Ltd., Yueyang 414000, China; cmliu803@sina.com

**Keywords:** bending, twinning, deformation mechanisms, texture, Mg-Al, magnesium alloys

## Abstract

The dependence of bendability on crystallographic orientations and texture evolution was investigated in a strongly textured Mg-9Al extrusion plate by bending along four directions. Results show that the bars have relatively small and reasonably close bendability when bent along the extrusion direction, transverse direction, and through-thickness direction. In contrast, the bendability of the 45° bar is much larger. Microstructure examination indicates that twins are prevalent in all bars. Furthermore, a detailed analysis of deformation mechanisms suggests that the initial texture transforms towards a basal texture during bending. Nevertheless, the texture transformation efficiency is drastically lower when basal slip—in contrast to tensile twinning—is the dominant deformation mechanism. The difference in texture evolution efficiency was used to rationalize the varied bendability along different directions. The findings of this provide insights into improving the bendability of magnesium alloys.

## 1. Introduction

Wrought magnesium alloys possess the most significant potential for weight-savings as structural components in the automotive and aerospace industries [1,2]. To reduce production costs, such parts are usually formed using sheet bending or similar processing at room temperature. However, magnesium alloys are known for their poor formability [3,4], which impedes their large-scale application.

Bending deformation involves complex stress states [5] that cannot be predicted simply from behaviors presented at uniaxial tension or compression. Various methods have been examined to explore routes for improving the bendability of magnesium alloys. Grain size refinement shows excellent potential since it benefits both yield stress and tensile ductility, yet bending studies show that decreasing grain size deteriorates the bendability of AZ31 [6,7] or provides no apparent improvement [8]. Micro-alloying with rare earth elements [9] and pre-processing [10,11,12,13] are reliable methods for improving bendability. Moreover, a further examination of these two methods shows that a weakened or altered texture is the key to enhanced bendability. Although materials with different compositions or texture features can bring valuable insights into a texture’s effect on bendability, sampling along different directions in one material avoids adulteration caused by materials and thus provides direct comparison and contrast.

To examine the effect of crystallographic orientation on bendability, an extrusion plate with a TD-TTD bi-component texture was bent along four directions. Assisted by examining texture evolution, this study will create a series of quantitative data from which a superior orientation can be selected to improve the bendability of magnesium alloys.

## 2. Experiments

The material used was an extrusion plate with a composition of Mg-8.8Al-0.47Zn-0.21Mn-0.17Ag (wt.%); other detailed information on this material can be found in our previous article [14]. Two sections were cut from the center of the plate: one was solution-treated at 420 °C for 1 h, and then water quenched to dissolve the Mg_17_Al_12_ phase formed during air cooling (referred to as S form hereafter); the other was first solution-treated and then aged at 175 °C for 40 h (referred to as A).

Bars for the bending test have a cross-section of 5 (width) × 3 (thickness) mm^2^ and a length no smaller than 32. These bars were machined from the heat-treated plate sections with the bar length along the extrusion direction (ED), transverse direction (TD), through-thickness direction (TTD), and 45° away from TTD in the thickness-transverse plane (45°). The bending tests were carried out on an Instron 3669 universal testing machine with a three-point bending test fixture with a 2 mm/min loading rate at room temperature. The loading and supporting pin radius was 10 mm, and the bending span was 30 mm. The sampling directions and illustration of the bending test are presented in Figure 1. To examine deformed microstructures, additional tests halted at a bending displacement of 3 mm and fracture (12 mm for the 45° bar due to the absence of fracture in the bending process) were also conducted with the same parameters. When the bars are bent to fracture, cracks occur and present as load drops in the displacement-load curves. By unloading at this point, a complete rupture can be avoided, and the microstructure features at the fracture thus can be preserved for characterization and analysis.

The texture information of the extruded plate and bent bars was characterized using electron backscatter diffraction (EBSD). The equipment used was a Helios Dual Beam field emission gun scanning electron microscope (SEM) equipped with an HKL Technology Channel 5.0 acquisition system. The samples for examination were wire-cut from the center and mechanically ground by 400, 1500, and 3500 grit sandpapers. Then, the samples were electro-polished in a solution of 96% ethanol + 1% perchloric acid + 3% Nitride acid under a voltage of 25 V at −40 °C for 2 min. The step size used in the scanning for the solution-treated condition and the bent bars were 3 and 0.7 μm, respectively.

## 3. Results

The aim of this study is twofold: (i) examining the effect of initial crystallographic orientations on bendability and (ii) characterizing the texture evolution features during bending. Both aspects of the aim can provide practical information on enhancing the bendability of magnesium alloys. In archiving this aim, initial orientations and bendability were correlated while the texture evolution process was compared and contrasted.

### 3.1. Initial Microstructure and Texture

Figure 2 presents the texture features of the extrusion plate in solution-treated conditions. The average grain size of the plate is determined to be 36 μm, and no difference in grain size is found along different directions when the plate is observed with an optical microscope. The texture of the extruded plate consists mainly of two components: the major one contains 60.5% of the grains, which have their ***c***-axes parallel to the TTD; the minor one contains 36.5% of the grains, which have their ***c***-axes parallel to the TD. Apart from these two main components, there are also about 3% of grains that have their ***c***-axes parallel to the ED. For the two main components, the spread angle between the ***c***-axis and the TTD and TD is around 30°, indicating a high sharpness of these components. Considering the high peak intensity (I_max_ = 15) of the major component, it can be concluded that the initial texture is both strong and sharp.

### 3.2. Bending Performance

Figure 3 shows the displacement-load bending response of the plate along four directions in both solution-treated and aged conditions. In the order of ED, TTD, TD, and 45°, the solution-treated bars show increased bendability. It is also noticeable that the ED, TD, and TTD bars show roughly the same level of mechanical performance; in contrast, the 45° bar displays significantly lower flow stress and larger bending displacement. To be exact, the 45° bar was not fractured after the bar was bent to an angle close to 180° under the same testing parameters. After aging, the four groups of bending bars present a similar pattern to that of solution-treated condition, with only a sharp decrease in displacement. These responses are consistent with the results acquired under unidirectional tension and compression tests with the same material [14].

### 3.3. Microstructure and Texture during Bending

Figure 4 shows the microstructures of the ED bar bent to a displacement of 3 mm and to fracture. When the ED bar is bent to a displacement of 3 mm, only a few tensile twins are observed in the grains with their ***c***-axis parallel to the ED on the tension side (Figure 4(1,2)). While on the compression side, profuse tensile twins invade nearly all the grains (Figure 4(3,4)). When the ED bar is bent to fracture, large amounts of double twins and a few contraction twins were observed on the tension side (Figure 4(5,6)). In contrast, on the compression side, tensile twins had nearly finished consuming all the grains, and double twins were found in the twinned regions with a similar frequency to that of the tension side (Figure 4(7,8)). It is worth noting that a few tensile twins occurred in grains with their ***c***-axis perpendicular to the tension direction (Figure 4(5)), and these twins might be caused by the spring-back during unloading.

Figure 5 and Figure 6 show the microstructures of the TD and TTD bar bent to a displacement of 3 mm and to fracture. The TD and TTD bars share similar microstructure features to that of the ED bar. At a displacement of 3 mm, only tensile twins are identified on both sides of the bar (Figure 5(2,4)), and the volume fraction of twins increases on the tension side while decreasing on the compression side in the order of ED, TD, and TTD. At fracture, remnant areas that are not invaded by tensile twins slightly increase on both sides of the TD and TTD bar than in the ED bar. More contraction twins were found on the tension side in TD and TTD bars than in ED. Furthermore, at the same time, double twins were found to be more prevalent than contraction twins on both sides in the TD and TTD bar.

Figure 7 shows the microstructures of the 45° bar bent to a displacement of 3 mm and to 12 mm. The 45° bar shows significant differences in microstructure from that of ED, TD, and TTD bars. In this bar, tensile twins were sparse on both sides when bent to a displacement of 3 mm (Figure 7(2,4)), and as indicated by the color code, the twins occurred in grains with relatively high Schmid factors (Figure 7(1,3)). When the bar was bent to a displacement of 12 mm, the tensile twinning was still an active deformation mechanism, especially on the tension side (Figure 7(6,8)). At the same time, contraction twins and double twins were also characterized on both sides of the bar (Figure 7(6,8)). However, these twins were activated on a much smaller scale than the ED, TD, and TTD bar.

## 4. Discussion

The extrusion plate’s bi-component configuration provides an excellent opportunity to examine a texture’s effect on bendability and texture evolution features. It is well-recognized that the side in contact with the indenter experiences compression while the side in contact with the supporting pins experiences tension. In the meantime, the extrusion plate has two main components allocated along TD and TTD (Figure 2). Bending the plate along four directions will thus put a series of orientation combinations under stress. When the tension and compression stress state are considered, a large set of results on a texture-bendability correlation and texture evolution can be acquired. These results provide crucial information on understanding and improving the bendability of magnesium alloys.

### 4.1. Dependence of Bendability on Grain Orientation

Although the bending process involves both tension and compression stress states, the bendability of a material is essentially similar to its ductility under a unidirectional tension or compression test: both are determined by the material’s metallurgical factors. For the four groups of bars, the difference in alloy composition can be neglected. The grains are equiaxed, and no differences were observed when viewed along different directions in optical microscopic examination. After solution treatment at 420 °C, a relatively high enough temperature for this alloy, precipitates would have been dissolved while residual dislocations or other microstructural defects were annihilated [15]. The only factor thus causing the difference in bendability in the solution-treated condition is the grains’ orientation, namely the initial crystallographic orientation.

### 4.2. Deformation Mechanisms and Their Effect on Texture Evolution

The combination of varied crystallographic orientations and complex stress states will inevitably cause the activation of various deformation mechanisms. With the help of a strong initial texture, the effect of these deformation mechanisms on texture evolution can be examined with higher precision.

#### 4.2.1. Tensile Twinning

Microstructure examination shows that {10$\overset{-}{1}$2} twinning (tensile twinning) is highly active during bending. As shown in Figure 2, the grains mainly have their ***c***-axis parallel to the TD and TTD, and Schmid factor analysis shows that the grains within the tilt angle of 30° from TD and TTD have a Schmid factor 0.48 for tensile twinning while having 0.19 and 0.20 for basal slip, respectively. When the CRSS ratio is considered [14], tensile twinning still has an advantage in activation priority.

Table 1 shows the volume fraction of the matrix that has been twinned by tensile twinning. In the compression side of the bent ED bar, the volume fraction of twins reaches 28% at a displacement of 3 mm while it is 95% at fracture. As shown in Figure 8a,b, the tensile twinning reorients the matrix in such a manner that the ***c***-axis is perpendicular to the tension direction while parallel to the compression direction. In the TD and TTD bars, the tensile twins occur at both tension and compression sides, although with relatively less frequency than the ED bar. It should be noted that tensile twinning was also observed in the 45° bar (Figure 8e,f), although the volume fraction was relatively low at a displacement of 3 mm and 12 mm.

#### 4.2.2. Contraction Twinning

At a fracture or a large displacement, contraction twins and double twins were characterized (Figure 4, Figure 5, Figure 6 and Figure 7) in all bent bars. Compared to the low frequency of contraction twins, double twins were much more prevalent. The difference in frequency is most likely due to the primary contraction twins being tensile-twinned as deformation continues [16]. Both the contraction twins and double twins confine themselves within areas no thicker than 2 μm. This thin cellular morphology and the reluctance to evolution originate from the high magnitude of atomic shuffles involved in propagating a small twin dislocation core and, thus, the low mobility of contraction twins [17]. As shown in Figure 8c,d,g, contraction twins, double twins, and recrystallized grains were formed in the matrix, and their ***c***-axes were scattered off the orientation of both matrix and loading stress. These new orientations contribute to the orientation randomization, although their low volume fraction is relatively low.

#### 4.2.3. Prismatic <a> Slip

Another primary deformation mechanism activated during bending is prismatic <a> slip (prismatic slip). As shown in Figure 2 and Figure 9b, the basal poles of most grains are distributed along the TD and TTD. From the perspective of the Schmid factor, these grains have a hard orientation for both tensile twinning and basal <a> slip (basal slip) under tensile load along ED, and in the meantime, these grains have a high SF for prismatic slip. To verify the activation of prismatic slip, the in-grain misorientation axis (IGMA) method [18,19,20] was adopted. Representative results are shown in Figure 10. The IGMA of the selected grains displays a concentration at the {0001} direction, indicating prismatic slip is the primary mechanism activated because no other mechanism in magnesium can cause such a pattern. The calculated contour line intensity of these grains exceeds 3.0, which is considered the benchmark value evidencing the activation of prismatic slip [18,19].

What should be clarified is that only some of the grains present a high concentration at the {0001} direction. This may originate from the counteractive effect between the high CRSS ratio between prismatic and basal slips [14] and the dynamic change of the CRSS ratio during deformation [21]. The former effect favors basal over prismatic slip, particularly at low deformation temperature, and thus explains the low frequency of grains with IMGA at {0001} direction. On the other hand, the CRSS ratio between prismatic and basal slips would decrease when the density of dislocation rises, and this effective CRSS ratio will exert positive feedback to the activation of prismatic slip, leading to extremely high intensity of these grains (Figure 10c).

#### 4.2.4. Basal Slip

For the 45° bar, the ***c***-axes of most grains are distributed well within a 15° deviation from the softest orientation (45°) relative to the stress direction. According to the Schmid factor law, a basal slip will be the dominant deformation mechanism due to this favorable orientation and its low CRSS. To validate the prediction based on the Schmid factor law, IMGA analysis is carried out. The result is presented in Figure 10. The IGMA of the grains 1, 2, 3, 5, 7, and 9 display a concentration mainly at the {$\overset{-}{1}$2$\overset{-}{1}$0} direction, while grains 6, 4, and 8 are at {01$\overset{-}{1}$0} or in between the {$\overset{-}{1}$2$\overset{-}{1}$0} and {01$\overset{-}{1}$0} direction. Most grains have a calculated contour line intensity surpassing 3, indicating that basal slip is only activated during deformation. The intensity spans in grains 4 and 8 suggest multiple basal slips were activated.

#### 4.2.5. Effect of Deformation Mechanisms on Texture Evolution

The activation of twinning or slip causes an abrupt or gradual shift of a grain’s orientation, respectively. When the grains activate the same deformation mechanism, texture evolves more rapidly than those in which multiple mechanisms are activated. Among these deformation mechanisms, tensile twinning contributes to texture evolution with the highest efficiency. As shown in the compression of the ED bar (Figure 11a), a low bending displacement caused a dramatic change in the texture configuration and peak intensity. As the bending displacement increased to fracture, nearly all the grains were reoriented, changing the initial {0001}//ED texture into a {0001}⊥ED texture. Drastic texture variation also occurred on both the tension and compression sides of the TD and TTD bars. The contraction twinning provided texture randomization (Figure 8c,d,g), but their low volume fraction and low twin boundary mobility rendered them far less efficient at altering the texture.

The prismatic slip is a vital mechanism to reorient the prismatic plane normally [22], yet it does not contribute to the rotation of the ***c***-axis [23]. Thus, no apparent changes were revealed by the {0001} pole figure in the tension side of the ED bar (Figure 11a). Basal slip is supposed to be the primary deformation mechanism during the bending of the 45° bar, but tensile twinning was activated simultaneously. The tensile twins formed under tensile stress led to texture randomization due to the variants’ different orientations, while the tensile twins formed under compression stress made nearly no contribution to texture evolution since the twins mainly shift between the two peaks due to the 86° re-orientation (Figure 8h). According to deformation geometry, a basal slip will cause the slip plane to rotate parallel to the tensile stress direction while perpendicular to the compressive stress direction [24]. As a result, the two peaks split from each other in tension while converging with each other in compression. The 5° deviation from the initial 45° in both tension and compression, on the other hand, demonstrates that basal slip is far less efficient in texture evolution compared to that on the compression side of the ED bar.

### 4.3. The Critical Role of Texture on Bendability

The alloys’ ability to resist bending loses rapidly as cracks initiate. Additionally, it has been well-established that crack initiation is deeply connected to contraction twinning and double twinning [25]. Although activation of contraction twinning has been found in alloys with a weak texture [16], large-scale activation of these twins will occur more readily when the ***c***-axis of the grains has been reoriented perpendicular to the loading axis in tension or parallel to the loading axis in compression. Thus, the initial texture affects bendability by its influence on texture evolution towards the formation of basal textures.

For the ED, TD, and TTD bars, basal textures are easily accessible on both the tension and compression side. In these bars, there is already a proportion of grains that have basal texture orientations before deformation. At the same time, the grains having other orientations can be easily reoriented towards a new basal texture relative to the stress direction by tensile twinning. In contrast to ED, TD, and TTD bars, no strong basal textures relative to the stress directions were formed in the 45° bar. To begin with, very few grains have their ***c***-axis parallel or perpendicular to the 45° direction before bending (Figure 2). Although grains near the peaks were able to activate tensile twinning in the late stage of deformation (Figure 7), the twins formed in tension only scattered away with the symmetrical axis being the TD or TTD, rather than a 45° direction, while at the same time, the twins formed in compression interchange between the TD and TTD peaks. Either way, no basal textures were able to form in the bending deformation of this bar. As for the basal slip, it can reorient the grains to form the basal textures, but the overall 5° migration in both the tension and compression side suggests that the basal slip is far less efficient in reorienting the basal plane.

The results of this study reveal that the key to increasing bendability is to enhance basal slip. This can be achieved in strongly textured materials such as the 45° sample in the present study: the combination of aggregated grain orientation and bending along the softest direction renders basal slip a dominant deformation mechanism for nearly all the grains, and thus a high bendability is acquired. However, this high bendability is hard to obtain in industrial practices since the dimension of the final product is usually limited in the thickness direction after rolling or extrusion. At the same time, bendability along other directions would be sacrificed due to the strong anisotropy attached to the strong texture. A more practical way is to weaken or randomize the basal texture produced during thermal–mechanical processing. Weak even no anisotropy would occur in this way, although the bendability will be relatively lower than the aforementioned methods since basal slip is not the dominant deformation mechanism for all grains.

## 5. Summary

An extruded Mg-9Al plate with a strong and sharp bi-component texture was bent along four directions to study the effect of crystallographic orientation on bendability. Deformation mechanisms activated during bending were examined, and their impact on texture evolution was analyzed. The following conclusions can be drawn:
The strongly-textured Mg-9Al plate has a relatively low bendability when bent along ED, TD, and TTD; when bent along the 45°, the plate’s bendability is dramatically enhanced.Bending along the three orthogonal directions activates extensive tensive twinning and prismatic slip while bending along the 45° direction activates mainly basal slip.The texture evolves uniformly towards the formation of basal textures when bent along all four directions: at the tension side of the bars, the basal plane is reoriented parallel to the direction of tensile stress, while on the inner side, the basal plane is reoriented perpendicular to the direction of compressive stress.The transforming efficiency towards the basal textures is high by {10$\overset{-}{1}$2} twinning and low by basal slip. The 45° bar’s high bendability can be explained by its efficiency in forming the basal textures during bending, and subsequently, the occurrence of contraction twins and fracture is postponed.This study shows that bendability can be improved by adjusting orientation, and the primary focus should be on those methods with which basal slip is enhanced.

## Figures and Tables

**Figure 1 materials-16-02518-f001:**
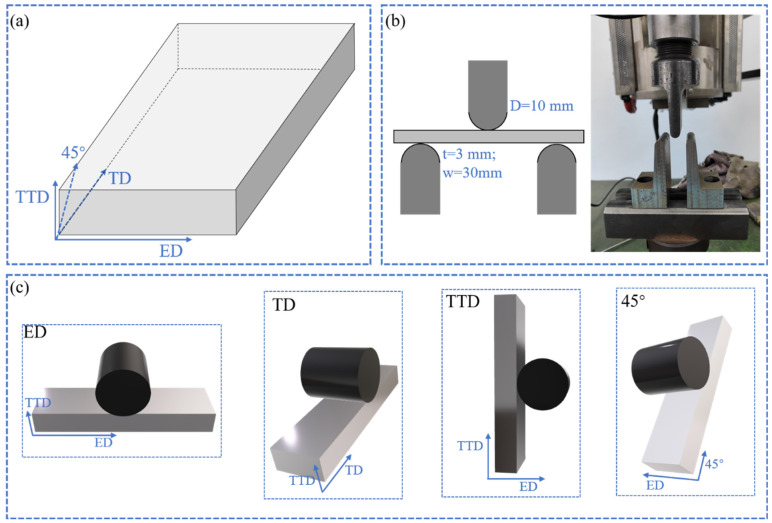
Diagram for sampling and bending test of the Mg-9Al extrusion plate: (**a**) sampling coordinate system; (**b**) schematic illustration and physical setup of the three-bending test; (**c**) coordinate system for the three-point bending test.

**Figure 2 materials-16-02518-f002:**
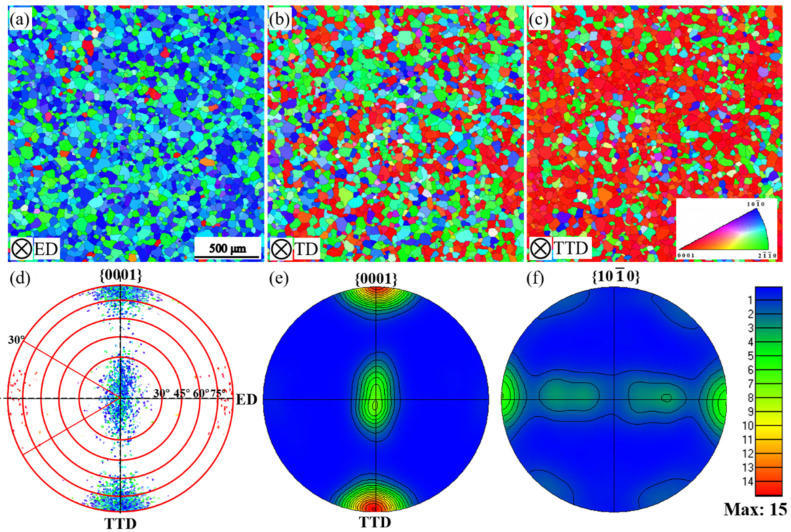
Microstructure and texture of the extruded Mg-9Al plate after solution treatment: (**a**–**c**) inverse pole figure maps viewed from ED, TD, and TTD, respectively; (**d**,**e**) {0001} pole figure in the scattered point and calculated contour line form; (**f**) {10$\overset{-}{1}$0} pole figure in the scattered point and calculated contour line form respectively.

**Figure 3 materials-16-02518-f003:**
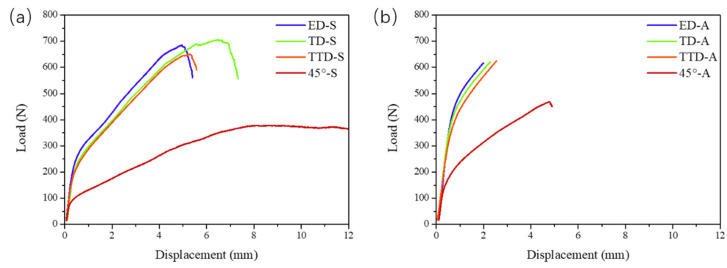
Mechanical response of the extruded Mg-9Al plate along four directions at (**a**) solution-treated condition; and (**b**) aged condition.

**Figure 4 materials-16-02518-f004:**
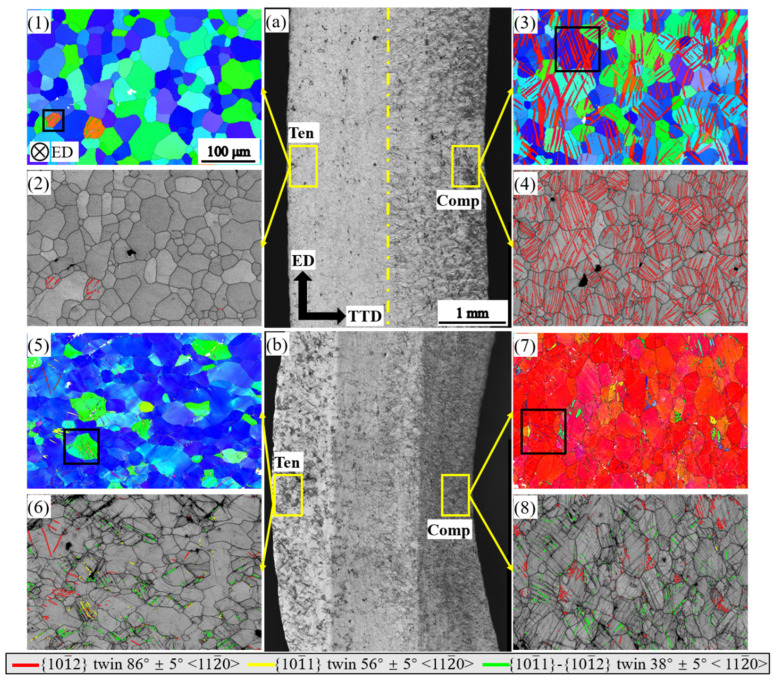
Microstructure of the ED bending bar at a bending displacement of 3 mm and at fracture: (**a**,**b**) optical micrograph of the bent sample; (**1**,**3**,**5**,**7**) inverse pole figures of the bent bar; (**2**,**4**,**6**,**8**) band contrast map of the bent bar with twinning boundary characterized.

**Figure 5 materials-16-02518-f005:**
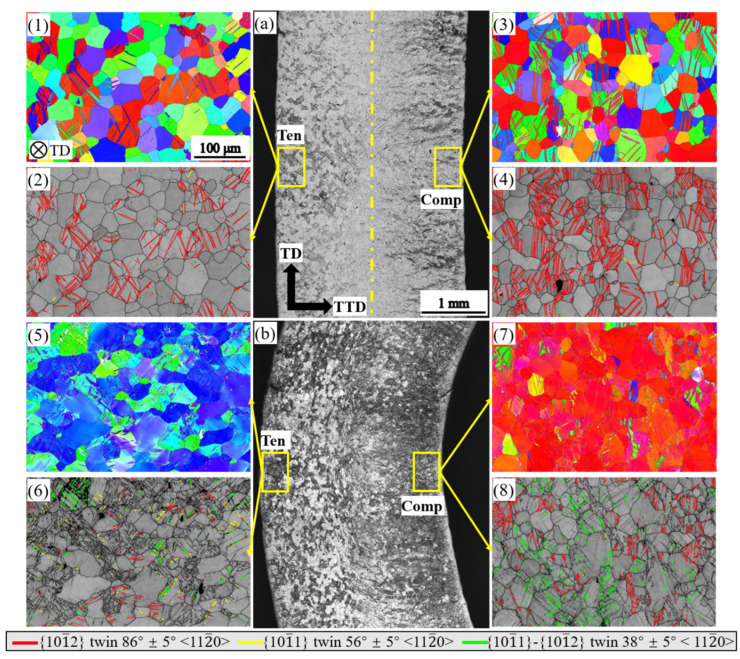
Microstructure of the TD bending bar at a bending displacement of 3 mm and at fracture: (**a**,**b**) optical micrograph of the bent sample; (**1**,**3**,**5**,**7**) inverse pole figures of the bent bar; (**2**,**4**,**6**,**8**) band contrast map of the bent bar with twinning boundary characterized.

**Figure 6 materials-16-02518-f006:**
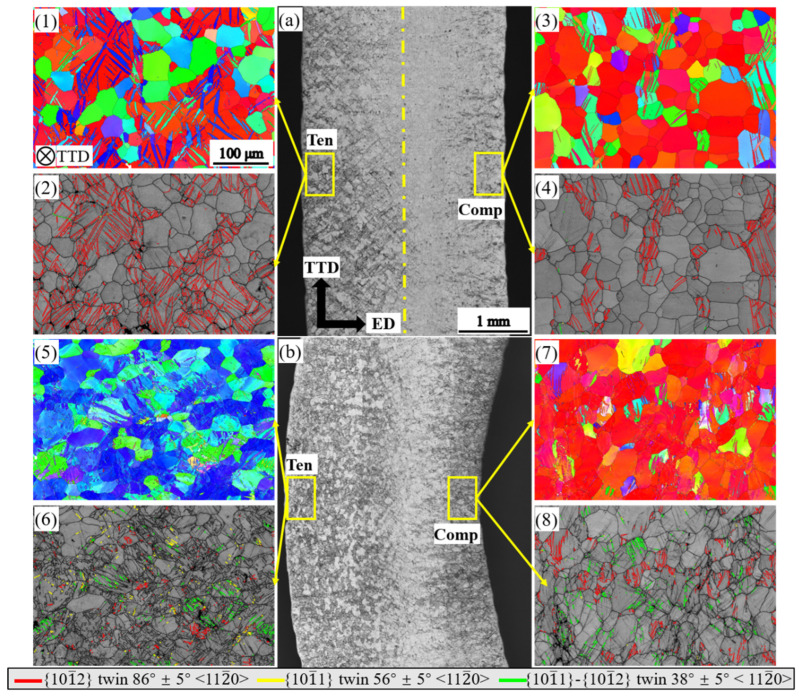
Microstructure of the TTD bending bar at a bending displacement of 3 mm and at fracture: (**a**,**b**) optical micrograph of the bent sample; (**1**,**3**,**5**,**7**) inverse pole figures of the bent bar; (**2**,**4**,**6**,**8**) band contrast map of the bent bar with twinning boundary characterized.

**Figure 7 materials-16-02518-f007:**
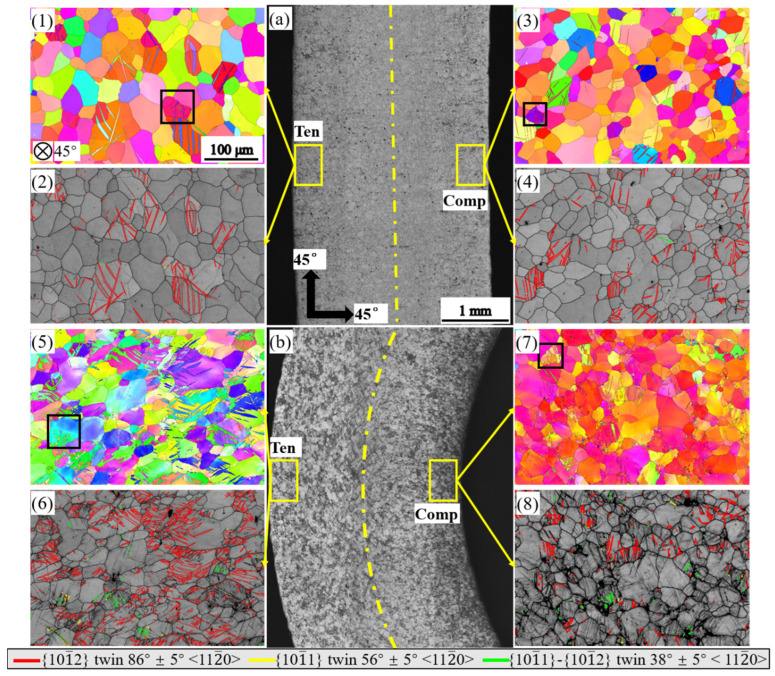
Microstructure of the 45° bending bar at a bending displacement of 3 mm and 12 mm: (**a**,**b**) optical micrograph of the bent sample; (**1**,**3**,**5**,**7**) inverse pole figures of the bent bar; (**2**,**4**,**6**,**8**) band contrast map of the bent bar with twinning boundary characterized.

**Figure 8 materials-16-02518-f008:**
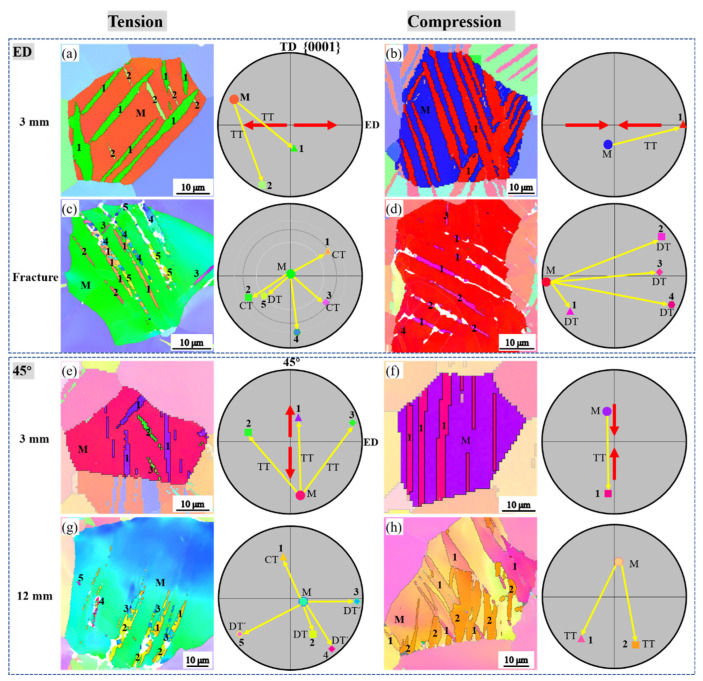
Illustration of orientation shift process caused by twinning in the bent ED and 45° bar: (**a**,**b**) tensile twinning in tension and compression at a bending displacement of 3 mm in ED bar; (**c**,**d**) contraction twinning and double twinning at fracture in ED bar; (**e**,**f**) tensile twinning at a bending displacement of 3 mm in 45° bar; (**g**,**h**) contraction twinning and double twinning at a bending displacement of 12 mm; all twinning types and variants identified were numbered (the selected grains have been marked with black rectangles in Figure 4, Figure 5, Figure 6 and Figure 7).

**Figure 9 materials-16-02518-f009:**
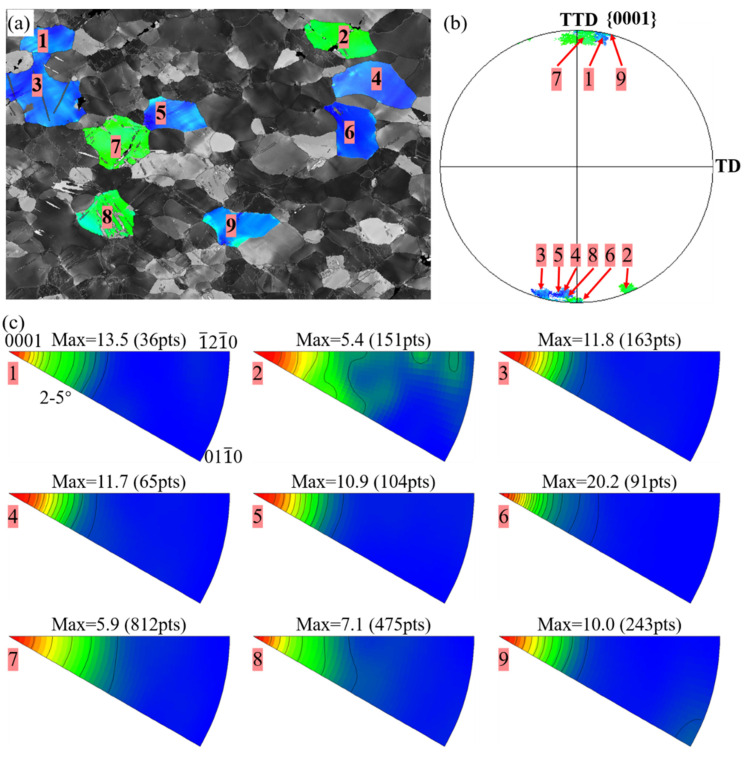
IGMA distribution for representative grains after bent ED bar: (**a**) inverse pole figure maps of the selected grains; (**b**) pole figures of the selected grains; (**c**) calculated IGMA of the selected grains.

**Figure 10 materials-16-02518-f010:**
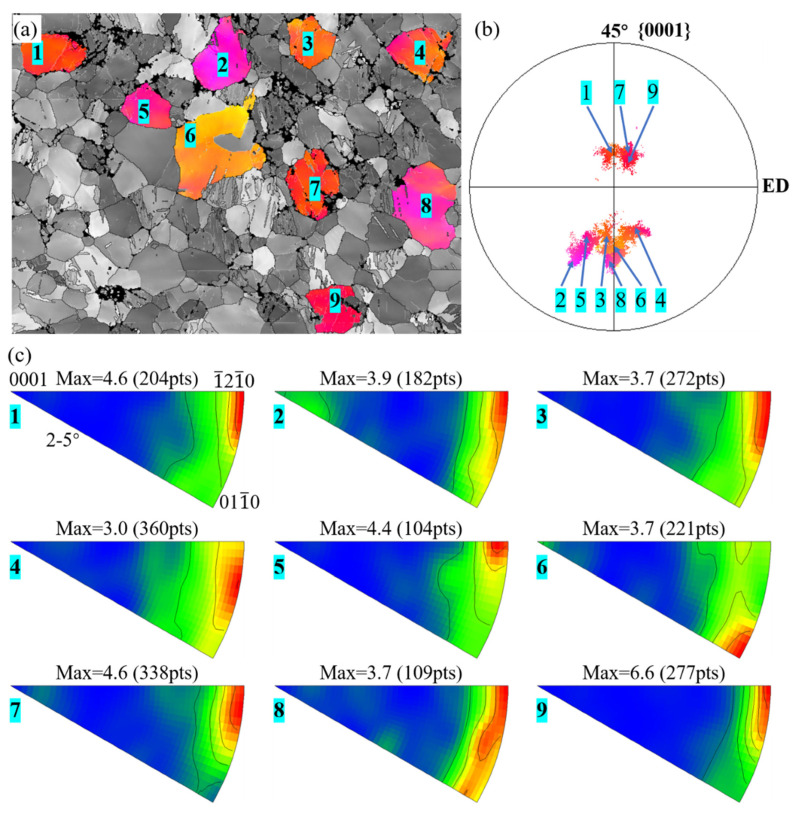
IGMA analysis of basal slip in the bent 45° bar: (**a**) inverse pole figure maps of the selected grains; (**b**) pole figures of the selected grains; (**c**) calculated IGMA of the selected grains.

**Figure 11 materials-16-02518-f011:**
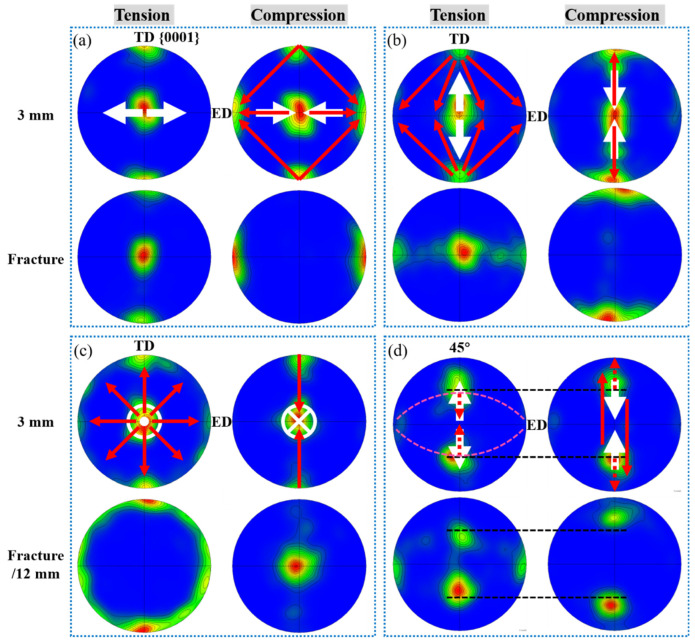
Texture evolution at the tension and compression side of the bar when bent to a displacement of 3 mm and fracture/12 mm along (**a**) ED, (**b**) TD, (**c**) TTD, and (**d**) 45° direction.

**Table 1 materials-16-02518-t001:** Volume fraction of matrix that has been reoriented by tensile twinning.

Direction	3 mm Ten	3 mm Comp	Fracture/12 mm Ten	Fracture/12 mm Comp
ED	0.5%	28%	3%	94%
TD	5.7%	14.7%	36.5%	55%
TTD	19%	8.4%	60.5%	25%
45°	5.8%	3%	20.7%	12%

## Data Availability

The raw data required to reproduce these findings are available from the corresponding author upon reasonable request.

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
