# Peer review of "Effect of Crystallographic Orientations on Bendability in a Strongly Textured Mg-9Al Extrusion Plate and Texture Evolution during Three-Point Bending"

_materials, 2023, doi:10.3390/ma16062518_

Round 1

Reviewer 1 Report

Dear authors,

it was my pleasure to read your work.

Unfortunately, the overwhelming number of references from chinese authors detracts from the otherwise very good work.

Best regards

Author Response

The authors can honestly claim that we did not pay particular attention to the authors’ nationality issue when we cited the references. The high percentage may have to do with the fact that many people are working in this field. We cited these works because the title and the keywords are relevant to the study we are working on, not because we know they are from Chinese people or names like Chinese people but with a different nationality. On the other hand, the authors admit that we are new to this particular field, and the choice of reference might not be perfectly sound.

Reviewer 2 Report

The topic of the article is not original as there are many articles already on this topic. However, the article brings a new perspective to the subject area that is worth considering. The conclusions contained in the article are consistent with the presented evidence and arguments and refer to the main purpose of the research. 

References are appropriate.

The article contains interesting research, well described and analyzed.

Author Response

Thanks for the recognition from the reviewer.

Reviewer 3 Report

The paper is good enough and deserves publication after a MINOR REVISION is made according to the following remarks.

1.      There is a misprint at the Introduction end. The authors mention a bi-component (initial) texture “TD-TD”, whereas it should be TD-TTD as evident from further explanations in Section 3.

2.      ED and TTD arrows are confused in the part of Fig. 1 for the “TTD bending” where the bar LENGTH should be parallel to TTD.

3.      When the semi-product (plate) is extruded, the best oriented bars (45deg orientation), limited by the plate thickness, prove to be rather short so that the length of final products would be also limited, somewhat devaluing practical significance of the obtained results. Appropriate reflections on this issue (at least, its mentioning) would be interesting to the reader.  

Author Response

(1) There is a misprint at the Introduction end. The authors mention a bi-component (initial) texture “TD-TD”, whereas it should be TD-TTD as evident from further explanations in Section 3.

Response:

The TD-TTD mistake has been corrected in the revised manuscript.

(2) ED and TTD arrows are confused in the part of Fig. 1 for the “TTD bending” where the bar LENGTH should be parallel to TTD.

Response:

The coordinate system of TTD bending has been corrected in the manuscript.

(3) When the semi-product (plate) is extruded, the best oriented bars (45deg orientation), limited by the plate thickness, prove to be rather short so that the length of final products would be also limited, somewhat devaluing practical significance of the obtained results. Appropriate reflections on this issue (at least, its mentioning) would be interesting to the reader.

Response:

The kind suggestion of the reviewer has been taken, and the final paragraph of the discussion has been adjusted accordingly in the revised manuscript.

Reviewer 4 Report

All the comments to the authors are in the attached file

Author Response

1) The manuscript is relatively easy to read. However, there are typos and phrases that could be changed for a better understanding of the work. I’m giving here just a few examples:

-Page 2 – line 55: “…length no smaller than 32…” I guess the authors are like to say 32 mm, please check it

-Page 4 – lines 99-104: “…In the order of ED, TTD, TD, and 45°, 99 the solution-treated
bars show increased fracture displacement…”; “…To be exact, the 45° bar was not
fractured after the bar was bent to an angle close to180° under the same testing
parameters…” . Even when is not wrong is a little confusing that the authors say in the same paragraph that samples goes to fracture and then not for one condition. I would recommend the authors to slightly change the sentence just to make it easy to understand.

- Page 12 – line 247: “… respectively. And when the…”. I think that is just a typo and should be without a point.

Response:

In the expression of “…length no smaller than 32…”, the authors are not saying the sample length is 32 mm for all samples. For the TTD sample, it is 32 mm; for the TD and ED sample is 50 mm since they are not limited by the plate dimension; while for the 45° sample, it is around 40 mm (smaller than 32*1.414 since the protruded edges will be grounded).  By using the expression of no smaller than 32 mm, the authors are trying to avoid the above-mentioned details and related explanation of why using that exact length, which the authors deemed are unnecessary details that may cause more questions from the readers like the one from the reviewer.

The authors are grateful for the reviewer’s kind suggestion, and the expression of “fracture displacement” has been replaced by “bending displacement”.

The and was removed in the revised manuscript so that the connection between the two sentences can be expressed with more separation, which is the original intention of the authors.

2) The authors show on the current manuscript results related to a previous work of Materials Characterization (reference 14). On section 2 Experiments the authors have decided not to include the material composition and use the reference. From my point of view, they should add to the composition just to allow the readers to easily find the data.

Response:

The authors are grateful for this suggestion, and information on the alloy composition has been added to the revised manuscript.

3) For this kind of texture evaluation regular reference system is ED-TD-ND instead of TTD, however is just an author choice, nevertheless, the axis in all the manuscript should be checked. In Figure 1 C the TTD sample shows ED-TTD axis and from my point of view there is an inversion of the axis between this one an Figure 1 a). All the axis should be clear for every condition, in Figure 2 a-c just one axis is shown, Figure 7 C the axis shown are 45-45. For texture analysis all the axis should be checked and carefully detailed.

Response:

The authors admit that there is mistake on the coordinate system of TTD bending, and it has been corrected in the revised manuscript. In Fig. 2 a-c, only one axis is needed to present the grain orientation and the EBSD observation plane was TD-TTD. In Fig. 7, however, two axes were adopted in order to present the exact observation plane (parallel to the length direction) of each bent bar. But the IPFs were selected in a viewing direction so that the twinning, especially tensile twinning can be observed more easily. As for the coordinate systems for the IPF and optical micrographs, the authors check carefully and we think those used in the revised manuscript are correct.

4) In section 3.1 the authors start to evaluate crystallographic texture of the material. Even when is not written, I assume that the (0001) pole figure is calculated from the EBSD data. When the authors talk about the two major texture components form the material these orientations are not shown in the regular (hkil)[uvtw] form. From the reviewer point of view this should be included and also and extra PF could help to clarify this point.

Response:

The {0001} pole figure was calculated from the EBSD data. The two components were not expressed in the regular form of (hkil)[uvtw] since the twinning behaviors and related bendability was mainly affected by the grains’ c-axis. To present more details of the grain orientation, {100} pole figure was added in the revised manuscript. 

5) The state on Section 4.1 that the “…difference in alloy composition can be
neglected…” I was assuming that during the whole manuscript the analyzed materials was the same; Am I correct on this point? Also, in this section authors state that …” After solution treatment at 420 °C, a relatively high 168 enough temperature for this alloy, precipitates would have been dissolved …” from the reviewer point of view to support this sentence an X-Ray diffraction pattern should be added or at least a reference where to check the dissolution temperature of the precipitation. Also, a reference is needed in line 175 where the authors say which one is the most prominent deformation mechanism.

Response:

Yes, the analyzed materials was the same extrusion plate used in a previous study, and the reason why the authors state this is that the alloy composition may fluctuate from position to position, but this fluctuation can be neglected in this study.

The suggestion of adding a reference to support the dissolution of precipitates was appreciated by the authors, and it has been followed in the revised manuscript.

In line 175, the authors meant to claim that the most prominent deformation mechanism is tensile twinning by the evidence provided by data in that paragraph. Since this may cause a misunderstanding, that claim was revised in the revised manuscript.  

6) On Figures 8, 9, 10 and 11 is not said which pole figure is shown, from my point of view is the (0001) but should be written. Also o Figure 11 not all the letters are written, is hard to know which one is Figure 11 e, i and so on.

Response:

The authors are grateful for this suggestion. Information on the {0001} pole figure has been added to the figures 8 to 11 in the revised manuscript. Also, the re-ordered indicators were assigned to the subset figures.

Reviewer 5 Report

1)     It is suggested that the title of the manuscript to have the following key details to reflect the main effort conducted by the authors

      “crystallographic orientations and texture evolution”

2)     Include preamble for section 3 to introduce its subsections.

3)     Include preamble for section 4.2 to introduce its subsections.

4)     Kindly include the physical setup of the experiment

5)     Caption/numbering of Subfigure of Figure 4 can be a bit confusing. Hence it is suggested that the subfigure (c) and (h) to become (a) and (b) whereas the zoom-in view can be numbered accordingly with roman numeral. Similar comment goes for Figure 5, Figure 6 and Figure 7.

6)     Kindly introduced Figure 4 until Figure 7 in the paragraph one at a time rather than introducing them altogether. (Referring to line 112)

7)     Explicit discussion on each of the subfigures in the Figure 4 until Figure 7 should also be included.

Author Response

1) It is suggested that the title of the manuscript to have the following key details to reflect the main effort conducted by the authors “crystallographic orientations and texture evolution”

Response:

The authors are grateful for this suggestion. The title has been revised accordingly.

2) Include preamble for section 3 to introduce its subsections.

Response:

The authors are grateful for this suggestion. A preamble was added for section 3.

3)     Include preamble for section 4.2 to introduce its subsections.

Response:

The authors are grateful for this suggestion. A preamble was added for section 4.2.

4)     Kindly include the physical setup of the experiment

Response:

The authors are grateful for this suggestion. A picture for the test setup has been added to the revised manuscript.

5)     Caption/numbering of Subfigure of Figure 4 can be a bit confusing. Hence it is suggested that the subfigure (c) and (h) to become (a) and (b) whereas the zoom-in view can be numbered accordingly with roman numeral. Similar comment goes for Figure 5, Figure 6 and Figure 7.

Response:

The authors are grateful for this suggestion, and the advice of the reviewer is followed exactly in the revised manuscript.  

6)     Kindly introduced Figure 4 until Figure 7 in the paragraph one at a time rather than introducing them altogether. (Referring to line 112)

Response:

The authors are grateful for this suggestion, and the advice of the reviewer is followed exactly in the revised manuscript.

7)     Explicit discussion on each of the subfigures in the Figure 4 until Figure 7 should also be included.

Response:

The authors are grateful for this suggestion. In some of the paragraphs, subfigures titles were added behind their corresponding description to ease the reading. In other paragraphs, no such modification was done, and the authors argue that more description may burden the readers unnecessarily.

Round 2

Reviewer 4 Report

Authors have improved the manuscript according to the referee´s suggestions and is ready to be accepted in the present form.